# SIN-Like Pathway Kinases Regulate the End of Mitosis in the Methylotrophic Yeast *Ogataea polymorpha*

**DOI:** 10.3390/cells11091519

**Published:** 2022-04-30

**Authors:** Hiromi Maekawa, Shen Jiangyan, Kaoru Takegawa, Gislene Pereira

**Affiliations:** 1Center for Promotion of International Education and Research, Faculty of Agriculture, Kyushu University, Fukuoka 819-0395, Japan; 2Graduate School of Bioresources and Bioenvironmental Sciences, Kyushu University, Fukuoka 819-0395, Japan; jy.shen96@outlook.com (S.J.); takegawa@agr.kyushu-u.ac.jp (K.T.); 3Faculty of Agriculture, Kyushu University, Fukuoka 819-0395, Japan; 4Center for Organismal Studies (COS), University of Heidelberg, 69120 Heidelberg, Germany; gislene.pereira@cos.uni-heidelberg.de; 5Division of Centrosomes and Cilia, German Cancer Research Centre (DKFZ), DKFZ-ZMBH Alliance, 69120 Heidelberg, Germany; 6Center for Molecular Biology (ZMBH), University of Heidelberg, 69120 Heidelberg, Germany

**Keywords:** mitotic exit network, septation initiation network, *Ogataea polymorpha*, Cdc15 kinase, SPB, Cdc5 kinase

## Abstract

The mitotic exit network (MEN) is a conserved signalling pathway essential for the termination of mitosis in the budding yeast *Saccharomyces cerevisiae*. All MEN components are highly conserved in the methylotrophic budding yeast *Ogataea polymorpha*, except for Cdc15 kinase. Instead, we identified two essential kinases OpHcd1 and OpHcd2 (*homologue candidate of ScCdc15*) that are homologous to SpSid1 and SpCdc7, respectively, components of the septation initiation network (SIN) of the fission yeast *Schizosaccharomyces pombe*. Conditional mutants for *OpHCD1* and *OpHCD2* exhibited significant delay in late anaphase and defective cell separation, suggesting that both genes have roles in mitotic exit and cytokinesis. Unlike Cdc15 in *S. cerevisiae*, the association of OpHcd1 and OpHcd2 with the yeast centrosomes (named spindle pole bodies, SPBs) is restricted to the SPB in the mother cell body. SPB localisation of OpHcd2 is regulated by the status of OpTem1 GTPase, while OpHcd1 requires the polo-like kinase OpCdc5 as well as active Tem1 to ensure the coordination of mitotic exit (ME) signalling and cell cycle progression. Our study suggests that the divergence of molecular mechanisms to control the ME-signalling pathway as well as the loss of Sid1/Hcd1 kinase in the MEN occurred relatively recently during the evolution of budding yeast.

## 1. Introduction

The exit from mitosis and the initiation of cytokinesis are important decision points in eukaryotic cell cycle regulation, particularly in budding yeast cell division where the site of cytokinesis (bud neck) is determined before spindle formation. In the budding yeast *Saccharomyces cerevisiae* and fission yeast *Schizosaccharomyces pombe*, the completion of mitosis is regulated by a GTPase-driven signalling pathway named mitotic exit network (MEN) and septation initiation network (SIN), respectively (hereafter collectively referred to as the mitotic exit (ME)-signalling pathway). The activation of MEN is essential for mitotic exit and cytokinesis, and SIN for cytokinesis [1,2]. In *S. cerevisiae*, ScCdc15 kinase mediates the activation of the GTPase ScTem1 to the downstream NDR kinase ScDbf2 complexed with its regulatory subunit ScMob1 [3,4,5] (Figure 1a).

Although MEN and SIN are evolutionally conserved pathways, some differences have been noted in the composition, regulation, and functional targets of the signalling components. For example, while ScCdc15 kinase directly activates the NDR kinase ScDbf2-ScMob1 in MEN, activation of the equivalent SIN NDR kinase SpSid2-SpMob1 requires the sequential action of two kinases: first, SpCdc7, which is the homologue of ScCdc15; second, SpSid1 complexed with SpCdc14 [8]. No Sid1 homologue has been reported in *S. cerevisiae*, *Ashbya gossypii*, or *Candida albicans*, which belong to the *Saccharomycetaceae* where the ME-signalling pathway has been investigated [9,10]. In contrast, the filamentous fungus *Aspergillus nidulans* has a SIN-like pathway to regulate septation [11].

In *S. cerevisiae*, MEN components are anchored to the spindle pole body (SPB), the equivalent of centrosomes in higher eukaryotes, and this association is significant in the regulation of the pathway [1,12,13]. The Tem1 GTPase together with the Bub2-Bfa1 GAP complex associates strongly with the daughter-directed SPB (dSPB) in metaphase and early anaphase [12,13,14,15]. The Cdc15 kinase is recruited to SPBs by the active Tem1 GTPase and activates Mob1-Dbf2 kinase at SPBs [16,17]. Active Mob1-Dbf2 kinase leaves the SPB and induces mitotic exit [18,19]. Importantly, the activation of MEN is regulated differently in the mother and daughter cell bodies [20,21,22,23,24,25,26]. Asymmetrically localising proteins at SPBs and the cell cortex play key roles to drive different regulatory mechanisms in each cell compartment.

The budding yeast *Ogataea polymorpha* has been used for industrial applications such as producing pharmaceuticals as well as being a subject of basic research related to its characteristics such as methylotrophy and thermotolerance. Recently, *O. polymorpha* has proved to be a valuable model system to study the evolutionary diversity of the molecular mechanisms involved in fundamental cellular processes such as the sexual cycle, cell cycle, and microtubule organisation [27,28]. Although SPB components and their regulation are thought to be largely conserved in *O. polymorpha* and *S. cerevisiae*, differences have been reported in the cell cycle regulation of the SPB structure and the role of OpCdc5 kinase [28]. These differences may represent the divergence of molecular regulations in the cell cycle in *O. polymorpha* from the well-studied model yeasts.

In this report, we identified two essential protein kinases, OpHcd1 and OpHcd2, as components of the conserved ME-signalling pathway in *O. polymorpha* (hereafter referred to as MES) with high sequence homology to Cdc7 and Sid1 in *S. pombe*. Localisation of the *O. polymorpha* MES components strongly suggest that MES activation occurs first at the mSPB and is prevented at the dSPB through OpBub2, which keeps OpTem1 in the inactive form. OpMob1-OpDbf2 kinase activated on the mSPB then moves to the dSPB, which is likely the key event to trigger mitotic exit. The close resemblance of the MES in *O. polymorpha* to the SIN in *S. pombe* suggests that mitotic regulation by the SIN-type pathway is the ancestral form in fungus and the changes to the MEN-type have occurred recently during the evolution of the budding yeasts.

## 2. Materials and Methods

### 2.1. Yeast Strains and Plasmids

Yeast strains and plasmids used in this study are listed in Appendix A. Unless otherwise indicated, all *O. polymorpha* strains were derived from NCYC 495 and were generated by PCR-based methods [29,30,31]. *O. polymorpha* cells were transformed by electroporation [32]. To generate *hcd1^M80G^* allele, primer OpHCD1_9 (ACCACCCAAAAACTCCCCAATAATCCACAGCTTGTATC) that encodes the M80G mutation was used to amplify the promoter region and codons 1–80 of *HCD1* ORF together with primer OpHCD1_7 (CGCTGCAGGTCGACGCGAGCATTTCGTCGATGAGG). The region containing a.a. 81–446 and the downstream terminator sequence was amplified with primers OpHCD1_10 (GAGTTTTTGGGTGGTGGATCC) and OpHCD1_8 (CTTAATTAACCCGGGACTTGCCGATCTCAGAGACC). These two DNA fragments were combined with BamHI (ER0054, Thermo Fisher Scientific Inc., Waltham, MA, USA) digested pFA6a-natNT2 plasmid using NEBuilder HiFi DNA Assembly Master Mix (E2621, New England Biolabs, Ipswich, MA, USA). The resulting plasmid (pHM1119) was digested with Nsi1 (ER0734, Thermo Fisher Scientific Inc.) and integrated at the *hcd1*Δ locus in HPH1737. The heterozygous diploid cells were transformed with Ku80+ plasmid (pHM898) and subjected to tetrad dissection to obtain a haploid clone carrying *hcd1::hphNT1<<Ophcd1^M80G^::natNT2*. Similarly, the *hcd2^L215G^* plasmid (pHM1121) was constructed with primers OpHCD2_7 (CGCTGCAGGTCGACGTTCCATGCGAACCACAGAAG), OpHCD2_8 (CTTAATTAACCCGGGCAATACGAAGACTAGCAGCC), OpHCD2_9 (ACTTTCGCAGTATTCGCCTATCAAATTCATAGACATCTCG), and OpHCD2_10 (GAATACTGCGAAAGTGGCTC). The pHM1121 DNA was digested with StuI (ER0421, Thermo Fisher Scientific Inc.), and used to transform HPH1738 by integration at the *hcd2Δ* locus. The heterozygous diploid cells were transformed with pHM898 and subjected to tetrad dissection to obtain a haploid clone carrying *hcd2Δ::hphNT1<<Ophcd2^L215G^::natNT2.* To construct iAID^Op^ plasmids, OsTIR-9myc, TetR, mAID, TetR-VP16 hybrid transactivator (tTA) gene-tetO7, spacer-5xflag, and hphNT1 fragments were PCR amplified from pNHK53, pST1760, pST1872 (obtained from NBRP Yeast), pCM225 [33], pKL260 (a kind gift from M. Kanemaki, National Institute of Genetics, Shizuoka, Japan), and pFA6a-hphNT1 [30], respectively. *OpURA3*, *OpLEU1*, *OpADE12* fragments including the ORF and 5′-upstream as well as 3′-downstream regions, *OpADH1* promoter, *OpTEF1* promoter and terminator, *OsSSN6* ORF (scaffold_1: 94041–96002) were amplified from *O. polymorpha* genome. These fragments were combined in pRS305 to generate iAID^Op^ plasmids.

To construct a Tet-OFF system for *O. polymorpha*, we replaced the *ScCYC* tata sequence and the *ScSSN6*-TetR’ (the reverse Tet repressor) with a tata-like sequence in the upstream region of *OpACT1* gene (Appendix A) and the *SSN6* homologue gene in *O. polymorpha* genome (scaffold_1: 94041–96002) fused with the TetR’, respectively. The TetR’ (the reverse Tet repressor)-*OpSSN6-flag* fusion gene was placed under the control of the strong constitutive *OpTEF1* promoter (P_TEF1_-TetR’-*OpSSN6-flag*) [27]. The reduction in protein expression was confirmed for *OpCDC5-flag* and *OpSPC72-flag* genes (Appendix A). To create the iAID^Op^ system, the *OpCDC5-flag* and *OpSPC72-flag* genes fused with mAID-tag were placed under the resulting Tet-OFF system and protein expression level was evaluated in various media conditions (Appendix A). The iAID^Op^ system depleted the mAID-Cdc5-flag protein by ~80% (Appendix A). The schematics and sequences of the plasmids used in this study are listed in Appendix A.

### 2.2. Yeast Growth Conditions and General Methods

Yeast strains were grown either in Yeast Extract–Peptone–Dextrose (YPD) medium (50 g/L Difco™ YPD Broth (BD242820, Becton Dickinson Co., Sparks, MD, USA)) containing 200 mg/L adenine (012-11512, Fujifilm Wako Pure Chemical Industries Ltd., Osaka, Japan), leucine (20327-62, Nacalai Tesque Inc., Kyoto, Japan), and uracil (212-00062, Wako) (YPDS) or in synthetic/defined (SD) medium (6.7 g/L Difco™ Yeast Nitrogen Base without amino acids, BD291940, Becton Dickinson Co.), 2% Glucose (64220-0601, Junsei Chemical Co. Ltd., Tokyo, Japan) supplemented with appropriate amino acids and nucleotides [34]. Cells were grown at 30 °C unless otherwise indicated. IAA (45533, Merck KGaA, Darmstadt, Germany), 5-Ad-IAA (A3390, Tokyo Chemical Industry Co., Tokyo, Japan), 5-Ph-IAA (30-003-10, BioAcademia, Osaka, Japan), were dissolved in ethanol for IAA, or DMSO (043-29355, Wako) for 5-Ad-IAA and 5-Ph-IAA to make a 500 mM stock solution and stored at −20 °C. To induce degradation of the endogenous protein fused with mAID, IAA, 5-Ad-IAA, or 5-Ph-IAA was added directly to the culture medium at the indicated concentration. Doxycycline (Z1311N, Takara Bio Inc., Shiga, Japan) was dissolved in H_2_O at 10 mg/mL and stored at −20 °C.

### 2.3. Microscopy

For the visualisation of DNA with 4′6,-diamidino-2-phenylindole (DAPI) (D9542, Sigma-Aldrich, Tokyo, Japan) in Figure 2 and Figure 3, yeast cells were fixed in 70% ethanol, washed with phosphate-buffered saline (PBS), and incubated in PBS containing 1 µg/mL DAPI. DAPI images were acquired using either an ECLIPSE Ti2-A inverted microscope (Nikon, Tokyo, Japan) equipped with a CFI Plan Apo Lambda 100 × objective lens (1.45 numerical aperture), a DS-Qi2 digital camera, an LED-DA/FI/TX-A triple band filter (Semrock: Exciter, FF01-378/474/575; Emission, FF01-432/523/702; Dichroic mirror, FF409/493/596-Di02), an LED light source X-LED1 and differential interference contrast (DIC), or BZ-700 with a PlanApo 60× objective lens (Keyence Co., Osaka, Japan). To observe GFP-Tubulin signal, Z-series images of 0.4 μm steps were captured using either a DeltaVision microscope (Applied Precision, Issaquah, WA, USA) equipped with GFP and TRITC filters (Chroma Technology Corp., Bellows Falls, VT, USA), a 100 × NA 1.4 UPlanSApo oil immersion objective (IX71; Olympus, Tokyo, Japan), and a CoolSNAP HQ camera (Roper Scientific, Trenton, NJ, USA) without fixation, or BZ-700 with a PlanApo 60× objective lens after fixing in 4% formaldehyde (063-04815, Wako) for 20 min. Images were analysed/processed with SoftWoRx 3.5.0 (Applied Precision, Issaquah, WA, USA), Priism4.3.0 software [35], or ImageJ 1.47 (NIH, Bethesda, MD, USA).

Live yeast cells carrying *OpHCD1-GFP*, *OpHCD2-GFP*, *OpTEM1-GFP*, *OpBFA1-GFP*, *OpMOB1-GFP*, or *MPS3-mRFP* were immediately analysed by fluorescence microscopy without washing or fixation. Z-series images of 0.4 μm steps were captured with DeltaVision equipped with a 60 × oil immersion objective (Olympus, Tokyo, Japan). Images were deconvolved and maximum intensity projections are shown. Time-lapse experiments were carried out in SD medium supplemented with adenine, leucine, and uracil on a glass-bottom dish (P35G-1.5-20-C, MatTek, Ashland, MA, USA) coated with concanavalin A (037-08771, Wako) at room temperature. Z-series at 0.4-μm steps were acquired every 90 s for Figure 4c,d, every 1 min for Figure 5c and Figure 6a,b, or every 2 min for Figure 7d. ImageJ 1.52 (NIH, Bethesda, MD, USA), Photoshop (Adobe Systems, San Jose, CA, USA), and Affinity Photo 1.8.4 (Serif (Europe) Ltd., Nottingham, UK) were used to produce merged colour images and assemble the image figures. No manipulations other than contrast and brightness adjustments were used.

### 2.4. RNA Analysis

Total RNA was isolated from *O. polymorpha* as previously described [27], treated with DNase I, and then further purified using the Monarch Total RNA Miniprep Kit (T2010S, New England Biolabs, Ipswich, MA, USA). A total of 250 ng RNA was used to synthesize cDNA with Reverse Tra Ace qPCR RT Master Mix (FSQ-301, Toyobo Co. Ltd., Osaka, Japan) according to the manufacturer’s protocol, and a 0.1–0.5 µL cDNA reaction mixture was used in qPCR reactions. Primers ACT1_8 (CTTCTTCCCAGTCTTCTGCTATC) and ACT1_9 (GGGCTCTGAATCTCTCATTACC) were used to amplify ACT1 RNA, and primers SPC72_Fw (ATGGCTGACCAAATCCTAGAC) and SPC72_Rv (GCTCTCAACTTTGCACTTAACC) for SPC72 RNA.

### 2.5. Yeast Cell Extracts and Immunoblotting

Whole cell extracts were prepared for SDS-PAGE and immunoblotting [30,36]. Samples representing 1–2 OD663 of a liquid culture were resuspended in 950 μL of cold 0.29 M NaOH (198-13765, Wako) and incubated on ice for 10 min. Then, 150 μL 55% (*w*/*v*) trichloroacetic acid (34637-85, Nacalai) was added and incubated for 10 min on ice. Protein pellets were collected by removing the supernatant after centrifugation at 20,817× *g* for 15 min at 4 °C, then resuspended in high urea buffer (8 M urea (35940-81, Nacalai), 5% SDS (313-90275, Wako), 200 mM NaPO_3_ pH 6.8 (169-04243 and 164-04295, Wako), 0.1 mM EDTA (311-90075, Wako), 100 mM dithiothreitol (042-29222, Wako), and bromophenol blue (05808-31, Nacalai) and heated at 65 °C for 10 min before loading on a gel. Western blotting was performed using a standard protocol. M2 monoclonal antibody (F1804, Sigma) was used to detect flag-tagged proteins.

## 3. Results

### 3.1. OpHCD1 and OpHCD2 Encode Kinases Similar to S. pombe SIN Kinases

A BLAST search of *O. polymorpha* genome sequence using *S. cerevisiae* MEN components (Tem1, Cdc15, Dbf2, Mob1, Lte1, Ste20, Bub2, and Bfa1) as query sequences identified homologues of all proteins except *ScCDC15* (Appendix A). Since ScCdc15 is a member of the Ste20 family of protein kinases, we suspected that one of the Ste20-like kinases may be a functional homologue and looked more closely at the hits in the BLAST search using the ScCdc15 amino acid query sequence. Among the top six hits, two ORFs had no obvious homologues in *S. cerevisiae* and were named *OpHCD1* and *OpHCD2* (*homologue candidate of ScCdc15*). We performed a phylogenetic analysis of OpHcd1, OpHcd2, and the four other top hits from *O. polymorpha* proteins, OpSte20 (scaffold_1:1061309–1063972), OpBck1 (scaffold_7:244619–245632), OpKic1 (scaffold_7:58132–582064), and OpSte11 (scaffold_3: 566290–568341); the proteins were named after the best-hit *S. cerevisiae* protein in BLAST search) along with the five proteins closest to either OpHcd1 or OpHcd2 in *S. cerevisiae* (ScSte20, ScSkm1, ScBck1, ScSte11, ScMkk2, ScSsk2, ScCdc15, ScSps1, ScKic1) as well as the eight closest hits in *S. pombe* (SpSte20, SpShk2, SpByr2, SpCdr1, SpCdc7, SpSid1, SpPpk11, SpNak1) (Figure 1b). The results showed that OpHcd1 and OpHcd2 have a similarity to ScCdc15, SpSid1, and SpCdc7. OpHcd2 displays 22% identity (35% similarity) to ScCdc15, and 21% identity (34% similarity) to SpCdc7. OpHcd2 has a protein kinase domain near the N-terminus, and in addition contains an armadillo type fold in the C-terminal region similar to SpCdc7 (Figure 1c). OpHcd1 is smaller in size (446 amino acids) and shows only 15% identity (22% similarity) to ScCdc15; however, the amino acid identity is higher to SpSid1 (38% identity and 55% similarity) (Figure 1c). Thus, these analyses suggested that OpHcd2 is the homologue to ScCdc15/SpCdc7 and OpHcd1 to SpSid1. Since a Sid1 homologue has not been reported in budding yeasts, we extended the BLAST search to other species in *Ascomycota*. Budding yeast species that diverged from the *S. cerevisiae* linage at early stages of evolution displayed an *OpHCD1*/*Spsid1* homologue gene in addition to the *ScCDC15* homologue in their genomes, while the *S. cerevisiae* linage has lost the *OpHCD1*/*Spsid1* homologue gene after the split with the *Wickerhamomyces* linage (Figure 1d). These results suggested that OpHcd1 and OpHcd2 are orthologs of SpCdc7 and SpSid1 and might play roles in late mitosis in *O. polymorpha*, and that the ancestral SIN-like signalling pathway has lost the Sid1 kinase relatively recently in budding yeast evolution.

### 3.2. OpHcd1 Plays Roles in Both Mitosis and Cytokinesis

To investigate the cellular functions of *OpHCD1* and *OpHCD2*, we first constructed deletion mutants. Heterozygous *hcd1*Δ*::hphNT1/HCD1* diploid cells were subjected to tetrad analysis, where the four spores of each ascus were analysed for the ability to grow on rich and selective plates. All asci formed one or two colonies (Appendix Aa). None of the growing cells were positive for the resistance marker (hygromycin) corresponding to the *hcd1*Δ*::hphNT1* deletion allele, while the segregations of heterologous auxotrophic *leu1-1* and *ura3-1* alleles were consistent with random segregation (*LEU1*:*leu1-1* = 8:2, *URA3*: *ura3-1* = 6:4 in 5 tetrads) (Appendix A). Similar results were obtained for heterozygous *hcd2*Δ*::natNT2/HCD2* diploid cells, where *HCD2* was deleted using nourseothricin (nat) resistance marker (Appendix A). In this case, no nat resistant colonies were obtained, while auxotrophic markers had close to random segregation (*LEU1*:*leu1-1* = 7:7, *URA3*: *ura3-1* = 6:8 in 7 tetrads). Thus, both *OpHCD1* and *OpHCD2* genes are essential for growth in *O. polymorpha*. Spores that were presumed to carry the *hcd1*Δ or *hcd2*Δ allele did not form micro-colonies after prolonged incubation. After germination, the number of cell bodies varied from spore to spore.

Construction of ATP-analog-sensitive alleles (as) is one of the best strategies to construct a conditional mutant allele for kinase encoding genes. In this strategy, the conserved “gatekeeper” residue in the ATP-binding pocket is mutated to glycine, enlarging the pocket so that cell permeable PP1 analogs, such as 1NM-PP1, can occupy it and selectively inhibit the kinase activity [37]. We introduced the as mutation into *OpHCD1* and *OpHCD2* according to the as allele of *ScCDC15* (Appendix A) [22]. The growth of *hcd1^M80G^* (hereafter called *hcd1-as*) but not *hcd2^L215G^* (hereafter called *hcd2-as*) cells was reduced on solid medium containing 0.5 µM 1NM-PP1, and almost abolished at 5 µM (Appendix A). Therefore, we initially proceeded with the phenotypical analysis of the *hcd1-as* allele of *OpHCD1*.

To investigate cellular functions of OpHcd1, logarithmically growing *hcd1-as* cells were treated with 1NM-PP1 and both cell division and cellular morphology were examined for up to two hours (Figure 2). Whereas the cell cycle duration of wild type cells was 1.5 ± 0.1 h, the mutants were unable to finish one cell cycle during the time of inspection (2 h, Figure 2b), implying prolonged delay or arrest of the cell cycle. The proportion of unbudded G1 cells was reduced, while that of large budded cells with two nuclei, corresponding to anaphase and telophase cells, was increased from 17.5% to 32.9% after one hour-incubation (Figure 2a), suggesting defects in late mitotic progression. After a two hour-incubation, the accumulation of cells that had initiated budding without completion of cytokinesis and/or cell separation became evident (Figure 2a,b). To clarify the defects, *hcd1-as* cells expressing *GFP-TUB1* (encoding for tubulin for spindle visualisation) were analysed. The proportion of *hcd1-as* cells with fully elongated anaphase spindles strongly increased after 2 h of 1NM-PP1 treatment compared to wild type cells (Figure 2c). Furthermore, some cells with a small bud were still attached to a neighbouring cell (Figure 2d). GFP-Tub1 patterns indicated that both the cell with the small bud and its neighbouring cell were in interphase. However, the cytoplasmic GFP signals were continuous between the two cells, suggesting that these cells were the mother and daughter cells from the previous cell cycle and that cytokinesis was still incomplete. These results suggested that OpHcd1 plays a role in mitotic exit as well as in cytokinesis and/or cell separation.

### 3.3. Optimisation of iAID System for Analysis of Hcd2 Function

In order to construct a conditional *hcd2* mutant, we next considered conditional depletion of OpHcd2 protein. Our first attempt to use the auxin-inducible degron (AID) system, which was successfully used for *OpCDC5* [28], did not generate a conditional mutant (Appendix A). We then attempted to establish a more efficient gene depletion method that is widely applicable in *O. polymorpha* by employing the improved AID system of *S. cerevisiae*, iAID, where the tetracycline(Tet)-OFF transcriptional repression system was combined with the existing AID system with a few necessary modifications. Since the Tet-OFF system of *S. cerevisiae* did not express model proteins in *O. polymorpha* (Appendix A), we first constructed the Tet-OFF system for *O. polymorpha* by introducing a tata-like sequence of *OpACT1* gene (TetO7-*OpACT1*tata, hereafter referred to as the TetO7 promoter) and *O. polymorpha* homologue gene of *ScSSN6* as described in the Materials and Methods, and then combined them with the AID system (hereafter referred to as iAID^Op^), and further improved by introducing the mutant versions of *OsTIR*, *OsTIR^F74G^*, and *OsTIR^F74A^*, that give higher sensitivity and specificity to IAA derivatives in various organisms including *S. cerevisiae* (Appendix A) [38].

Application of the iAID^Op^ system to the *OpHCD2* gene was successful in generating a conditional iAID^Op^-*hcd2-as* allele (Figure 3a). The *Os**TIR^F74A^*, version of the iAID^Op^-*hcd2-as* allele was used for further analysis since it conferred a tighter growth-defective phenotype in the presence of ≥1 µM 5-Ph-IAA or 5-Ad-IAA compared to wild type *OsTIR* with 500 µM IAA (Figure 3a and Appendix A).

### 3.4. OpHcd2 Functions in Mitotic Exit and Cytokinesis

We next analysed the effect of OpHcd2 depletion to obtain an insight into the OpHcd2 function (Figure 3b,c). Upon OpHcd2 depletion, the percentage of unbudded G1 cells decreased and remained low, suggesting that the cell cycle was delayed or arrested, while it did not change significantly in wild type cells during the same period (Figure 3c, upper graphs). After 1 h of depletion and inhibition of mAID-Hcd2-as protein, anaphase/telophase cells (large budded cells with two segregated nuclei) increased to 58% in iAID^Op^*-hcd2-as* but not in wild type cells (Figure 3c, lower graphs). Longer incubation times did not increase the proportion of anaphase/telophase cells, but instead led to the appearance of cells with more than three cell bodies that remained connected (Figure 3c, purple), indicating cytokinesis defects. To confirm this, wild type and iAID^Op^*-hcd2-as* cells expressing GFP-Tub1 were analysed. In iAID^Op^*-hcd2-as* mutants, the proportion of cells containing anaphase spindles increased 1 h after the depletion of mAID-Hcd2-as protein but did not exceed 50% (Figure 3d,e). On further incubation up to 2 h in total, the proportion of anaphase cells slightly decreased and chains that had three or more unseparated cell bodies accumulated (Figure 3e). Close inspection of cytoplasmic GFP signals in such cells revealed that the cytoplasm between the cell bodies was often continuous (Figure 3f). Thus, these results suggested that the depletion of OpHcd2 delayed mitotic exit as well as cytokinesis and/or cell separation.

### 3.5. OpHcd1 and OpHcd2 Preferentially Associate with the SPB in the Mother Cell Body in Late Anaphase

One of the characteristics of the MEN/SIN is the SPB localisation of the core components during mitosis [1]. To investigate whether OpHcd1 and OpHcd2 are SPB-associated kinases, we first examined their intracellular localisation using cells carrying *OpMPS3-mRFP* as an SPB marker [28]. Both OpHcd1-GFP and OpHcd2-GFP signals were observed at the SPB that resides in the mother cell but only in late anaphase cells (Figure 4a,b). Time-lapse microscopy revealed that OpHcd1-GFP and OpHcd2-GFP signals appeared at the SPB residing in the mother cell only after spindle elongation in anaphase (Figure 4c, 9 min and Figure 4d, 9 min). The OpHcd2-GFP SPB signal became weak and disappeared around the time of mitotic exit (Figure 4c, 16.5 min), while the Hcd1-GFP signal remained at the SPB until the 30 min time point, before it then gradually decreased during the next G1 after the 31.5 time point (Figure 4c–e). Importantly, Hcd1-GFP and Hcd2-GFP were never observed at the SPB in the bud in either single image captures or time-lapse experiments. Together, these results suggest that OpHcd1 and OpHcd2 are recruited to the mSPB only in mid/late anaphase and their recruitment and maintenance at the SPB are regulated differently in the mother and daughter cell compartment.

### 3.6. SPB Association of Conserved MEN Core Components Are Regulated Spatially and Temporally during the Cell Cycle of O. polymorpha

The SPB localisation of OpHcd1 and OpHcd2 was similar to that of ScCdc15 only in its timing, but differed spatially as both *O. polymorpha* kinases bind only to the SPB in the mother cell body. This prompted us to ask about the cellular localisation of other core components of the MES in *O. polymorpha*. In *S. cerevisiae*, the most upstream Tem1 GTPase and its negative regulator Bub2-Bfa1 complex associate with SPBs from G1 to metaphase and then strongly and preferentially with the SPB in the bud during anaphase [13]. While ScTem1 signals on both SPBs become nearly equal before the end of anaphase, the ScBub2-ScBfa1 complex signal remains asymmetric throughout the cell cycle [13]. Similarly, in *O. polymorpha*, the SPB signal of OpBfa1-GFP exhibited strong asymmetry in which a signal appeared only at the SPB in the bud, although a faint signal was observed at the mSPB in some late anaphase cells (Figure 5a,b). Time-lapse microscopy revealed that the bud-directed SPB accumulates OpBfa1-GFP signal more strongly during anaphase (Figure 5c, the 15–25 min). The OpBfa1-GFP asymmetry became strong in mid-anaphase when the dSPB entered the bud (15 min) and persisted until the end of mitosis (25 min). Establishment of the cytoplasmic microtubules–bud cortex interaction, which ensures continuous movement of the SPB into the bud, may determine the timing of OpBfa1 accumulation at the dSPB. In contrast to OpBfa1, we observed a continuous SPB association of OpTem1 that started in early anaphase (Figure 6a, 12 min) and remained at both SPBs until the time of mitotic exit (Figure 6a, prior to 31 min). No apparent asymmetry of OpTem1-GFP signal was observed during mitosis. Next, we examined Mob1 localisation in time-lapse experiments. OpMob1-GFP signal appeared at the mSPB in anaphase and persisted until the end of mitosis (Figure 6b, 27 min). The dSPB lacked OpMob1-GFP for most of anaphase but acquired OpMob1-GFP before the end of mitosis (Figure 6b, 17 min). So the duration time of OpMob1-GFP signal during anaphase was shorter at the dSPB than the mSPB (Figure 6c). Together, our data show that MES components associate with the SPBs in *O. polymorpha*, indicating conservation of the ME-signalling components between *O. polymorpha* and both *S. cerevisiae* and *S. pombe*.

We also examined the localisation of the conserved Cdc14 phosphatase in a time-lapse experiment in *O. polymorpha*. Similar to ScCdc14 in *S. cerevisiae*, OpCdc14-GFP was also observed as nucleolus-like foci in interphase cells, which became dispersed first to the nucleoplasm as cells entered anaphase and later into the cytoplasm. At the end of mitosis, OpCdc14-GFP reformed as strong foci in each nucleus (Figure 6d, 30–38 min). These results suggest that OpCdc14 phosphatase might also be regulated at the level of nucleolar retention as in *S. cerevisiae*.

### 3.7. Bub2 Inhibits SPB Association of Hcd1 and Hcd2 in Daughter Cells

The asymmetry of OpHcd1 and OpHcd2 localisation at SPBs may represent the sequential activation timing of the MES in *O. polymorpha*, first in the mother cell body and then later in the bud. To obtain insight into this asymmetry, we first examined OpHcd2 and OpHcd1 localisation in *bub2*∆ cells since Tem1 GTPase is prematurely active in metaphase in *bub2*∆ or *bfa1*∆ cells in *S. cerevisiae* [13,15,17,39]. In the absence of OpBub2, OpHcd2 prematurely appeared at the SPBs in the majority of cells at G1, S, and G2 phase as well as in the majority of metaphase cells, with 80% of the cells having OpHcd2-GFP at both SPBs (Figure 7a,b). Furthermore, cells expressing the constitutive active allele of *TEM1*, *OpTEM1^E70L^*, which we generated based on the *ScTEM1^Q79L^* allele [40], exhibited similar OpHcd2-GFP localisation to that in *bub2*∆ cells (Figure 7a,b). In *OpTEM1^E70L^* cells, the SPB association of OpHcd2-GFP occurred at early stages of the cell cycle and was symmetric in its distribution on both SPBs after SPB separation (Figure 7a asterisks, Figure 7b). These results suggest that OpHcd2 is recruited to SPBs when OpTem1 is activated, while OpBub2 prevents OpHcd2 localisation to the dSPB. Similarly, OpHcd1-GFP localisation was symmetric in anaphase *bub2*∆ cells (Figure 7c,d). However, in contrast to OpHcd2, with OpHcd1 in *bub2*∆ cells only ~50% of the G1 cells had OpHcd1-GFP localized at SPBs, and there was no premature localisation to SPBs in S/G2 or metaphase (Figure 7d). We reasoned that the SPB association of OpHcd1 in *bub2*∆ G1 cells might result from delayed dissociation from SPBs after mitotic exit of the preceding cell cycle.

Together, our data indicate that OpBub2 may play a significant role in inhibiting the MES at the dSPB by preventing SPB binding of OpHcd2 and OpHcd1 in anaphase. Our data also suggest that the SPB association of OpHcd1 is restricted to anaphase by a mechanism that requires a mitotic regulator other than active Tem1.

### 3.8. Polo-Like Mitotic Kinase Cdc5 Restricts the SPB Association of Hcd1 to Anaphase

To identify Tem1-independent regulators of OpHcd1 SPB binding, we next examined the involvement of the polo-like kinase OpCdc5 in OpHcd1 for two reasons. First, OpCdc5 associates with SPBs in mitosis and, second, the *S. cerevisiae* Cdc5 has been shown to control ScCdc15 localisation independently of ScTem1 activation [16,28]. Auxin-induced depletion of OpCdc5 resulted in the accumulation of anaphase cells with segregated SPBs in the mother and the daughter cell bodies [28] (Figure 8). In the majority of wild type or OpCdc5-depleted anaphase cells, Hcd2-GFP localised to only one SPB, suggesting that OpCdc5 has little or no role in inhibiting the SPB binding of OpHcd2 to both SPBs similar to hyperactive Tem1 (Figure 8a). In contrast, SPB localisation of OpHcd1-GFP was almost completely abolished in OpCdc5 depleted cells (Figure 8b,c). The small number of cells that did carry SPB signals was probably caused by the incomplete degradation of OpCdc5 protein in some of the *OpCDC5-AID* cells. These results indicate that the mitotic kinase OpCdc5 is essential for recruiting OpHcd1 to SPBs during anaphase in *O. polymorpha*.

Taken together, all of the localisation studies suggest that OpHcd2 is mainly regulated by the Tem1 status, while OpHcd1 requires the mitotic Cdc5 kinase in addition to Tem1 activation for SPB binding (Figure 8d).

## 4. Discussion

In this study, we investigated the ME-signalling pathway in *O. polymorpha* and obtained what is likely to be the complete list of core components consisting of Tem1, Bub2-Bfa1, Hcd1, Hcd2, Mob1-Dbf2, and the downstream target Cdc14. The SPB localisations of these core *O. polymorpha* components revealed diversity in the spatial regulation of the ME-signalling pathway in budding yeast species. We propose that the MES in *O. polymorpha* is more similar to the SIN in *S. pombe* than the MEN in *S. cerevisiae*.

### 4.1. OpHcd1 and OpHcd2 Are Required for Mitotic Exit and Cytokinesis

Our phenotypical analysis of *hcd1-as* and iAID^Op^*-hcd2-as* mutants as well as the amino acid sequence conservation of OpHcd1 and OpHcd2 to SpSid1 and SpCdc7, respectively, indicate that OpHcd1 and OpHcd2 play important roles in mitotic exit and cytokinesis and thus could be considered as likely components of an MEN/SIN-homologous signalling pathway in *O. polymorpha*. Because *hcd1-as* and iAID^Op^*-hcd2-as* cells only transiently arrested the cell cycle in anaphase during time-course experiments, we assume that these conditional mutant alleles have some leakiness in liquid media, even though they exhibited a severe growth defect on solid medium under restrictive conditions. It is unclear why the analogue-sensitive mutation in the *OpHCD2* gene was insufficient for the construction of a conditional mutant (Appendix A). However, it is known that cells can tolerate the reduced activity of several kinases [37], and lower levels of Hcd2 activity may be sufficient for cell division.

While the SIN in *S. pombe* primarily regulates cytokinesis, the MEN in *S. cerevisiae* is essential for mitotic exit and cytokinesis [41]. Similarly to the MEN, the pathway in *C. albicans* plays key roles in driving mitotic exit, cytokinesis, and cell separation [10,42], and mutant phenotypes of the core components suggest their distinct roles are needed to achieve the cellular functions of the ME-signalling pathway: CaTem1 and CaCdc15 for mitotic exit, CaDbf2 primarily for cytokinesis, and the non-essential CaCdc14 for cell separation [42,43]. The results of this study further enforce the notion that the regulation of mitotic exit was placed under the ME-signalling pathway during the evolution of Saccharomycetaceae.

### 4.2. ME-Signalling Pathway Senses Tem1 Activity and Cell Cycle Phase

The MEN in *S. cerevisiae* and the SIN in *S. pombe* are equivalent signalling pathways. However, they are divergent in their signalling architecture and the role of each component in the regulation of mitotic exit and cytokinesis [40]. While the activation of the ScTem1 GTPase in the MEN is transduced to the most downstream NDR kinase ScDbf2-ScMob1 through a single kinase ScCdc15, the SIN requires two kinases. Phylogenetic evidence clearly indicates that the SIN type is the ancestral form and the loss of the second kinase occurred more recently in budding yeast evolution (Figure 1d and Figure 8d). The significance of having the second kinase in the pathway is not understood. The MEN-type signalling may simply have altered the regulatory mechanism to compensate for the loss of the second kinase. However, it is possible that the MEN-type have an advantage over the SIN-type for the budding style of cell division. Importantly, ScCdc15 is a simultaneous detector for both Tem1 activation, which occurs when one nucleus enters into the bud (spatial), and the Cdc5 kinase activity, which indicates the cell is in the mitotic phase (temporal), to ensure that mitotic exit occurs only after the completion of nuclear division and segregation [16]. Our study on the MES in *O. polymorpha* revealed that the two kinases have different roles in coordinating the activation of the downstream kinase with the completion of mitotic events. We propose that OpHcd2 kinase (SpCdc7-like kinase) senses Tem1 activation, while OpHcd1 kinase (SpSid1-like kinase) is under the control of OpCdc5 (Figure 8d, the purple and red thick arrows). OpHcd1 might, thus, ensure that the MES activation is transduced to the downstream targets only after cells have entered anaphase. It may be reasonable to speculate that Cdc15 in *S. cerevisiae* inherited functions from both of the two kinases in the SIN-like pathway and became a merging point of different signals that had separate target kinases in the SIN-type. Further study in *O. polymorpha* on the molecular details of how OpHcd2 and OpHcd1 kinases sense Tem1 and Cdc5 activities will provide insight into the conservation and diversity of the spatial and temporal regulation in fungi in general.

### 4.3. Asymmetric SPB Localisation of the ME-Signalling Pathway in O. polymorpha

Our study indicated that the MES in *O. polymorpha* is an SPB-associating pathway and the core components have their own preference for which SPB they bind (Figure 8d). We propose that in *O. polymorpha* the Hcd1 and Hcd2 activation steps might start in the mother cell compartment, which is supported by two observations. Firstly, OpHcd1 and OpHcd2 localise preferentially to the mSPB (Figure 4). Secondly, disturbance of the asymmetric OpHcd2 and OpHcd1 localisation in *bub2*Δ cells indicates that OpBub2-OpBfa1 negatively regulates Tem1 in anaphase at dSPB, while OpTem1 is activated at the mSPB, which does not carry OpBub2-OpBfa1 in *O. polymorpha* (Figure 7). Preferential dSPB binding of OpBub2-OpBfa1 and/or its selective removal from the mSPB likely contribute to restrict MES activation to the mSPB (Figure 8d, anaphase). In contrast to Hcd1 and Hcd2, OpMob1 appears at the dSPB as well as the mSPB shortly before mitotic exit, suggesting that the recruitment of the active Mob1-Dbf2 kinase complex to the dSPB may trigger mitotic exit. The molecular details of MES activation remain largely elusive, e.g., where OpTem1 activation initiates, what specifies the preferential binding of OpHcd1 and OpHcd2 to mSPBs, and how OpMob1 (probably in a complex with Dbf2) is recruited and activated at dSPBs. If the dynamic localisation of OpTem1 at the SPBs is similar to that of ScTem1, then this may allow it to escape inhibition at the dSPB by OpBub2-OpBfa1 and allow its activation in the vicinity of the SPB in the bud. The active Tem1 may travel to the mother cell compartment where it can transduce the signal to the downstream Hcd1 and Hcd2 kinases and then the most downstream OpMob1-OpDbf2 kinase at mSPBs. Active OpMob1-OpDbf2 leaves the mSPB to move into the bud and then be recruited to the dSPB. Another possibility is that active OpHcd1 and/or OpHcd2 moves into the bud and activates OpMob1-OpDbf2 in cytoplasm, which results in recruiting OpMob1-OpDbf2 to the dSPB. Although the relationship between OpHcd1 and OpHcd2 is unclear, we assume OpHcd1′s SPB localisation is either downstream or independent of OpHcd2 because OpHcd1, but not OpHcd2, requires OpCdc5 (Figure 8).

Preferential SPB association is a common feature of MEN and SIN components, such as ScTem1/SpSpg1 GTPase and the ScBub2-ScBfa1/SpByr4-SpCdc16 GAP complex. However, whether the asymmetry is functionally significant in MEN/SIN regulation remains controversial. In *S. cerevisiae*, based on numerous studies on protein localisation and the protein dynamics of MEN components, two alternative models, the “zone model” and the “sink model”, have been proposed [13,21,22,24,40,44,45,46,47,48,49,50,51,52]. The zone model proposes ScTem1 is activated at the dSPB when the SPB leaves the mother cell compartment (inhibitory zone) and enters into the bud (activating zone), while the sink model proposes that the strong ScBub2-ScBfa1 asymmetry restricts Tem1 inhibition to the dSPB and allows Tem1 activation at the mSPB. Recently, Campbell et al. [53] proposed that MEN activation is controlled differently at the two SPBs and active MEN at both SPBs is required for the timely exit from mitosis. Our model for the asymmetric regulation of the MES in *O. polymorpha* predicts similar mechanisms to those proposed for the MEN in *S. cerevisiae* (Figure 8d). ScTem1 is activated at the dSPB and recruits ScCdc15 kinase to be activated at the dSPB. Active ScCdc15 then binds both SPBs [53]. Similarly, we assume that in *O. polymorpha*, OpTem1 activation initiates at the dSPB because we did not observe SPB localisation of OpHcd1 or OpHcd2 in early anaphase when both SPBs remain in the mother cell body [28]. ScMob1, which localises equally to both SPBs in late anaphase, first associates with the mSPB in early anaphase [47]. This might reflect the activation of OpMob1 at the mSPB first and then propagation of the signal to the dSPB in *O. polymorpha*. Regulation for strong asymmetry of OpHcd1 and OpHcd2 localisation to the mSPB does not resemble that for ScCdc15. It is interesting to note that SpCdc7 and SpSid1 bind to only one SPB during anaphase in *S. pombe* (Figure 8d) [54]. Thus, one part of the MES regulation in *O. polymorpha* is similar to the MEN, while another part is similar to the SIN, which suggests the MES in *O. polymorpha* may represent the intermediate form during the evolutionary change from the SIN-type pathway to the MEN. Homologues of the SPB outer plaque components, ScNud1 and ScSpc72, that tether the MEN are conserved in *O. polymorpha*, but whether and how these SPB proteins are involved in the MES regulation has not been addressed [28]. Such studies will provide new insights into the asymmetric SPB localisation of the ME-signalling components and their conservation among yeast species.

## Figures and Tables

**Figure 1 cells-11-01519-f001:**
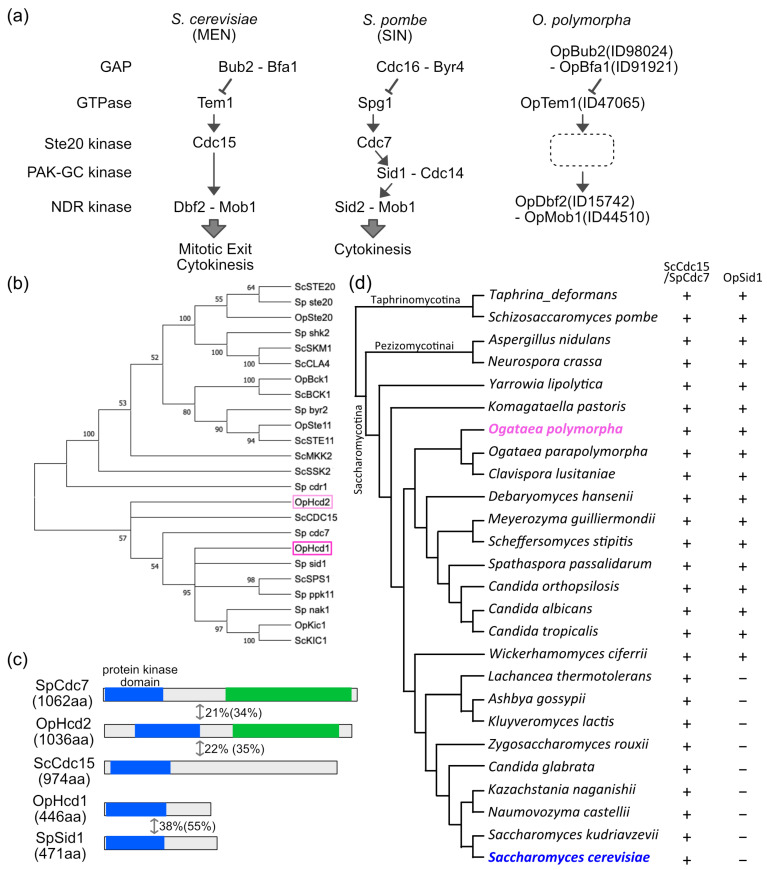
*OpHCD1* and *OpHCD2* encode protein kinases homologous to Sid1 and Cdc7 in *S. pombe*. (**a**) Schematics of the core components of the MEN and SIN and the homologous pathway in *O. polymorpha*. The protein IDs for *O. polymorpha* proteins are according to reference [6]. (**b**) Phylogenetic tree of protein kinases from *S. cerevisiae*, *O. polymorpha*, and *S. pombe* that have similarity with *S. cerevisiae* Cdc15. Phylogenetic tree was constructed in MEGAX with the Maximum Likelihood Method. Shown is the Bootstrap Consensus Tree. (**c**) Schematic representation of the OpHcd1, OpHcd2, ScCdc15, SpCdc7, and SpSid1 proteins. Blue and green boxes are the protein kinase domain and the armadillo type fold, respectively. The numbers are the amino acid identity and the similarity (in bracket) from the alignment between two kinases across the length of the proteins. (**d**) Conservation of ScCdc15/SpCdc7 and SpSid1 in Ascomycota. Phylogenetic relationships are based on Shen et al. [7] Tree is not in scale.

**Figure 2 cells-11-01519-f002:**
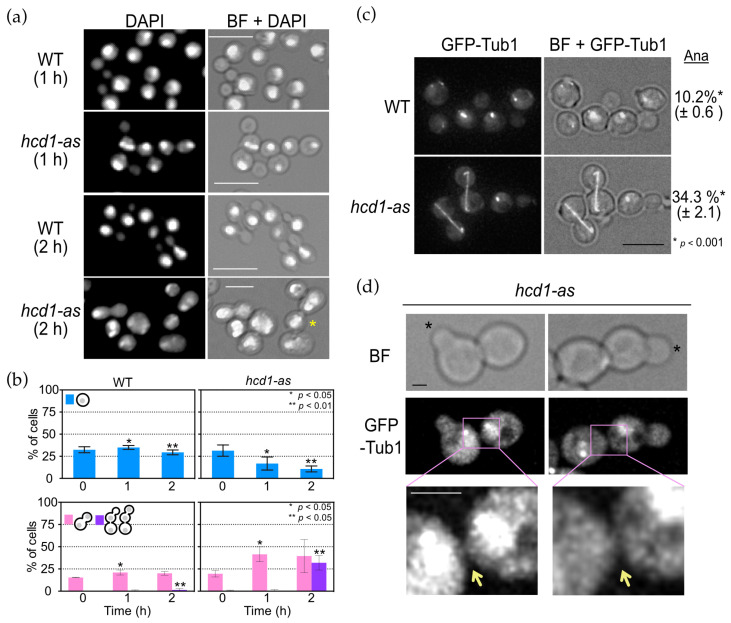
*Hcd1-as* cells have defects in mitosis and cytokinesis. (**a**) Logarithmically growing wild type (HPH1047) and *hcd1-as* (HPH1894) cells in YPDS medium were incubated with 5 µM 1NM-PP1 at 30 °C for the indicated time. Cells were fixed with 70% ethanol and DNA was stained with DAPI. BF, brightfield. Scale bar, 10 µm. Asterisks indicate unseparated cells. Scale bar, 10 µm. Note that cells are larger in *hcd1-as* (2 h). (**b**) Quantification of a. Upper graph: percentage of unbudded cells with one nucleus. Lower graph: percentage of large budded cells with two nuclei, one in the mother and the other in the bud, and unseparated large cell bodies with two or more nuclei. More than 100 cells were analysed at each time point. The experiment was performed in triplicate and the combined results are shown. Statistical significance of the differences between wild type and *hcd1-as* cells was determined by *t*-test and indicated by single or double asterisk. (**c**) Logarithmically growing wild type (HPH1968) and *hcd1-as* (HPH1969) cells carrying *GTP-TUB1* in YPDS medium were incubated with 5 µM 1NM-PP1 at 30 °C for 1 h. Images were captured without fixation. Shown are projected images. BF, brightfield. Scale bar, 5 µm. Statistical significance was determined by the *t*-test. (**d**) Incomplete cytokinesis and cell separation of *hcd1-as* cells from the experiment in c. Asterisks indicate newly formed buds. Yellow arrows point to the connections between mother and daughter cells of the last cell cycle. BF, brightfield. Scale bar, 1 µm.

**Figure 3 cells-11-01519-f003:**
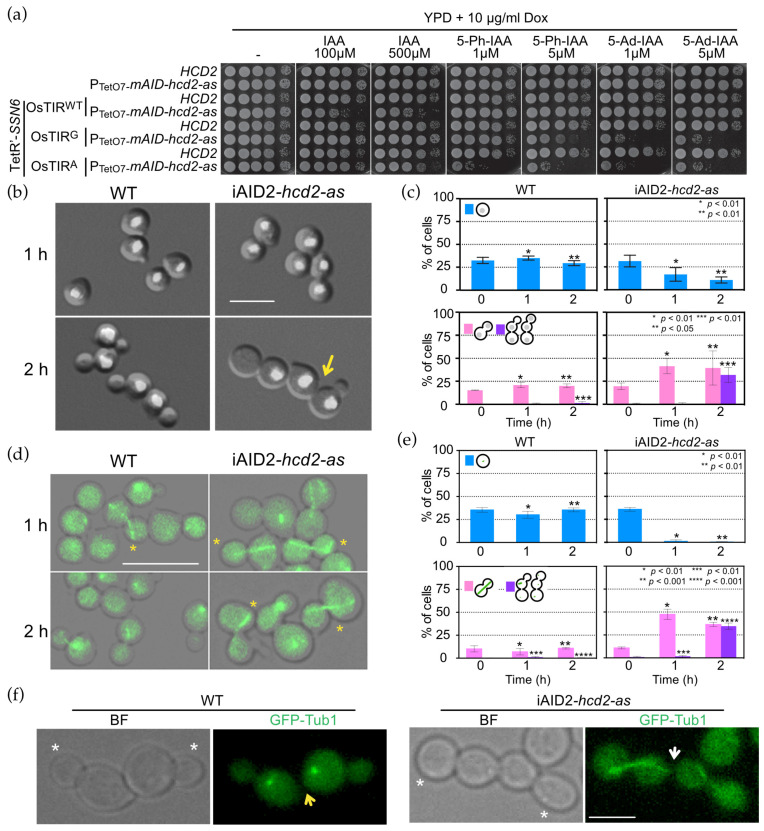
OpHcd2 depleted cells are defective in mitotic exit and cytokinesis. (**a**) Serial dilutions of strains with the indicated genotype were spotted on YPDS agar plates containing IAA, 5-Ph-IAA, or 5-Ad-IAA at the indicated concentrations and incubated at 30 °C for 1 day. Yeast strains: HPH656, HPH2067, HPH2254, HPH2246, HPH2270, HPH2244, HPH2245, HPH2247. (**b**) DAPI staining of iAID^Op^*-hcd2-as* strains. Wild type (HPH2247) and iAID^Op^*-hcd2-as* cells (HPH2245) carrying P_ADH1_-*OsTIR^F74A^*, P_CMV_-tTA, P_OpTEF1_-*TetR’-OpSSN6-5flag* were grown in YPDS medium until logarithmic growth phase. Dox was added at 5 µg/mL to repress the expression of the *mAID-hcd2-as* gene, IAA at 0.5 mM to induce the mAID-OpHcd2-as protein degradation, and 1NM-PP1 at 5 µM to inhibit the activity of any residual iAID^Op^-Hcd2-as that escaped degradation, and incubated at 30 °C for up to 2 h. Cells were fixed with 70% ethanol and DNA was stained with DAPI. Shown are merged images of brightfield and DAPI images. Scale bar, 5 µm. Arrows indicate failure of cell separation in the previous mitosis. (**c**) Quantification of b. Upper graphs: percentage of unbudded cells with one nucleus. Lower graphs: percentage of large budded cells with two nuclei, one in the mother and the other in the bud, and three or more unseparated cell bodies with two or more nuclei. Statistical significance was determined by the *t*-test and is indicated by asterisks. (**d**) GFP-tubulin was examined by epifluorescence microscopy in wild type and iAID^Op^*-hcd2-as* strains. Wild type (HPH2258) and iAID^Op^*-hcd2-as* cells (HPH2260) carrying P_ADH1_- *OsTIR^F74A^*, P_CMV_-tTA, P_OpTEF1_-*TetR’-OpSSN6-5flag* and expressing *GFP-TUB1* were grown in YPDS medium until logarithmic growth phase. Dox, IAA, and 1NM-PP1 were added at 5 µg/mL, 0.5 mM, and 5 µM, respectively, and incubated at 30 °C for up to 2 h. Cells were fixed with 4% formaldehyde for 20 min and then washed with PBS before microscopy. Asterisks indicate anaphase spindle. Shown are merged images of brightfield image and DAPI images. Scale bar, 5 µm. (**e**) Quantification of d. Upper graphs: percentage of unbudded G1 cells. Lower graphs: percentage of budded cells with anaphase spindle and three or more unseparated cell bodies without anaphase spindle. Statistical significance was determined by the *t*-test and indicated by asterisks. (**f**) Incomplete cytokinesis and cell separation of iAID^Op^-*hcd2-as* cells from the experiment in (**d**). Asterisks indicate newly formed buds. Yellow and white arrows point to the complete separation and the connection between mother and daughter cells of the previous cell cycle, respectively. Brightfield (left) and epifluorescence (GFP) (right) image. Scale bar, 2 µm.

**Figure 4 cells-11-01519-f004:**
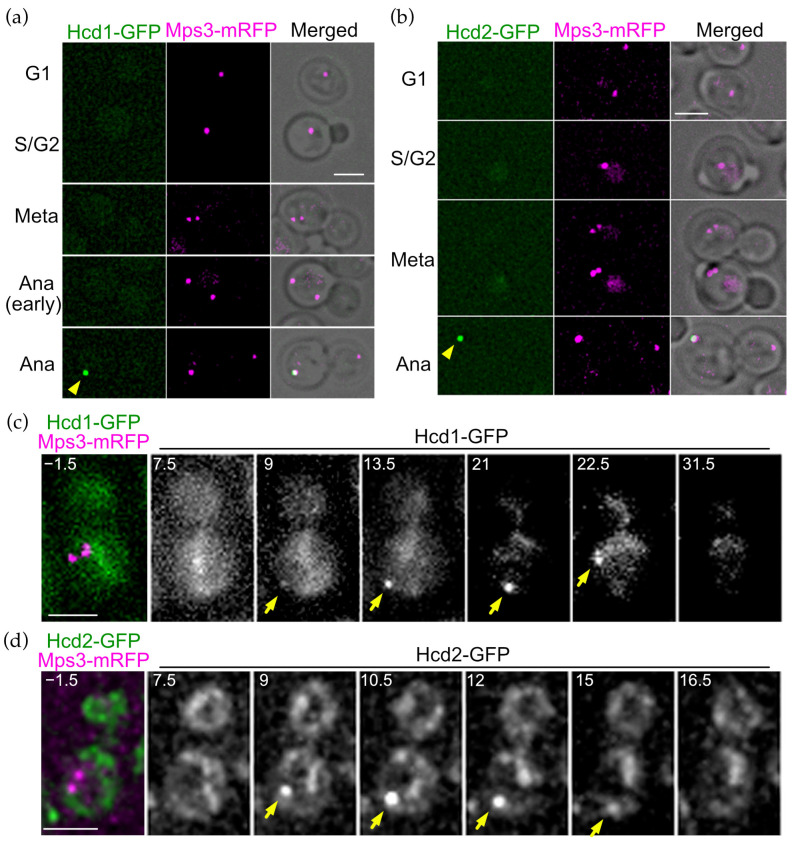
OpHcd1 and OpHcd2 localise to the SPB in the mother cell body during anaphase. (**a**) Cell cycle dependent localisation of OpHcd1-GFP. *OpHCD1-GFP MPS3-mRFP* cells (HPH1605) were grown in YPDS medium at 30 °C. Mps3-mRFP is an SPB marker. Shown are deconvolved and projected GFP and RFP images and merged brightfield, GFP, and RFP images. Yellow arrowhead indicates Hcd1-GFP signal. Meta: metaphase. Ana: anaphase. Scale bar, 2 µm. (**b**) Cell cycle dependent localisation of OpHcd2-GFP. *OpHCD2-GFP MPS3-mRFP* cells (HPH1608) were grown in YPDS medium at 30 °C. Mps3-mRFP is an SPB marker. Shown are deconvolved and projected GFP and RFP images and merged brightfield, GFP, and RFP images. Yellow arrowhead indicates Hcd2-GFP signal. Meta: metaphase. Ana: anaphase. Scale bar, 2 µm. (**c**) Time-lapse microscopy of *OpHCD1-GFP MPS3-mRFP* cells (HPH1605). Images were taken every 90 sec. RFP image was only captured before the start of the time-lapse series. The cell entered into anaphase before the 9 min timepoint judged by the position of the Hcd1-GFP dot. The Hcd1-GFP signal returned to the cell centre at the 22.5 timepoint, suggesting that mitotic exit occurred between the 21 min and the 22.5 min timepoints. Yellow arrowheads indicate the OpHcd1-GFP signal at the mSPB. Shown are deconvolved and projected images. Scale bar, 2 µm. (**d**) Time-lapse microscopy of *OpHCD2-GFP MPS3-mRFP* cells (HPH1608). Images were taken every 90 sec. RFP image was only captured before the start of the time-lapse series. The cell entered into anaphase before the 9 min timepoint as judged by the position of the Hcd2-GFP dot. Yellow arrowheads indicate the OpHcd2-GFP signal at the mSPB. Shown are deconvolved and projected images. Scale bar, 2 µm.

**Figure 5 cells-11-01519-f005:**
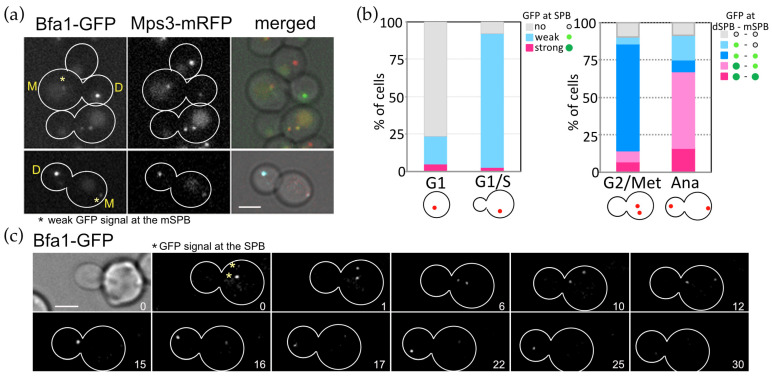
Asymmetric SPB binding of OpBfa1 in the cell cycle. (**a**) Cell cycle dependent localisation of OpBfa1-GFP. *OpBFA1-GFP MPS3-mRFP* cells (HPH1761) were grown in YPDS medium at 30 °C. Mps3-mRFP is a marker for the SPB. Shown are deconvolved and projected GFP and RFP images and merged brightfield, GFP, and RFP images. Scale bar, 2 µm. Yellow M and D mark the mother and daughter cell compartments of a large budded cell, respectively. Asterisks indicate a weak GFP signal at the mSPB. (**b**) Quantification of cells in a. Cells were categorised based on budding and the number and position of the SPB. GFP signals were judged as no, weak, or strong signals. N > 200 cells. In quantification of G2/Metaphase cells, the SPB locating at the position closest to the bud was designated as dSPB and the other SPB as mSPB. (**c**) Time-lapse microscopy of *OpBFA1-GFP MPS3-mRFP* cells (HPH1761). Images were taken every 1 min. Only GFP signal was captured. Yellow asterisks indicate metaphase SPBs at the start of the time series. The cell entered into anaphase between the 10 min and 11 min timepoints. Scale bar, 2 µm.

**Figure 6 cells-11-01519-f006:**
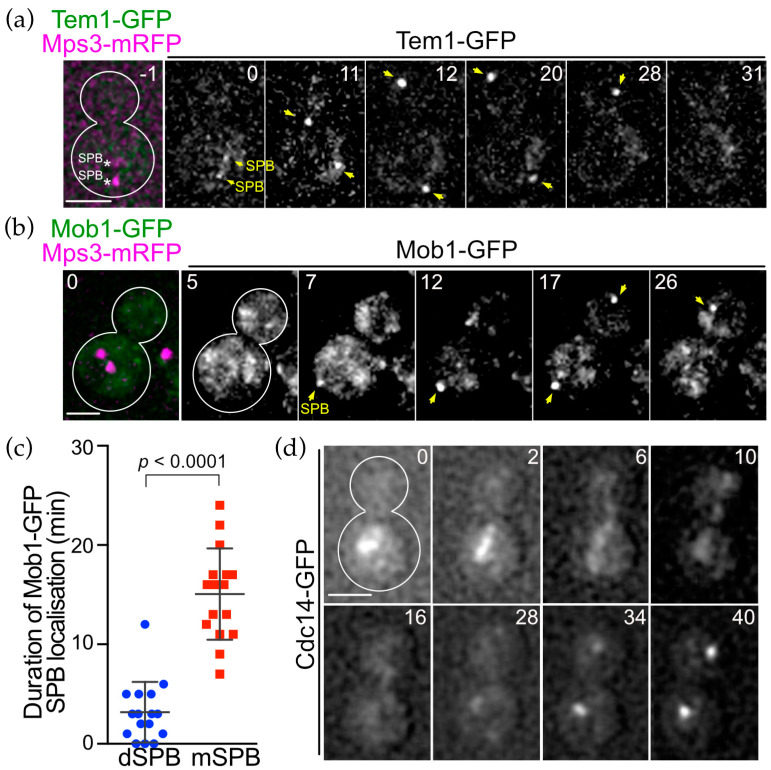
SPB association of OpTem1 and OpMob1 is cell cycle dependent. (**a**) Time-lapse microscopy of *OpTEM1-GFP OpMPS3-mRFP* cells (HPH1771). Images were taken every 1 min. The RFP image was only captured before the start of the time-lapse series. Anaphase onset was around the 12 min timepoint. The dSPB was inserted into the bud at the 15 min timepoint. Yellow arrows indicate SPBs. Scale bar, 2 µm. (**b**) Time-lapse microscopy of *OpMOB1-GFP OpMPS3-mRFP* cells (HPH2108). Images were taken every 1 min. The RFP image was only captured at the first timepoint. The cell entered into anaphase before the 7 min timepoint. The dSPB was first detected at the 17 min timepoint. Yellow arrows indicate SPBs. Scale bar, 2 µm. (**c**) Duration time of Mob1-GFP at the mSPB and dSPB was measured in the time-lapse experiment in (**b**). N = 16. Statistical significance was determined by the *t*-test. (**d**) Time-lapse microscopy of *OpCDC14-GFP* cells (HPH1479). Images were taken every 2 min. The cell entered into anaphase shortly before the 2 min timepoint. The anaphase nucleus was inserted into the bud before the 6 min timepoint. Scale bar, 2 µm.

**Figure 7 cells-11-01519-f007:**
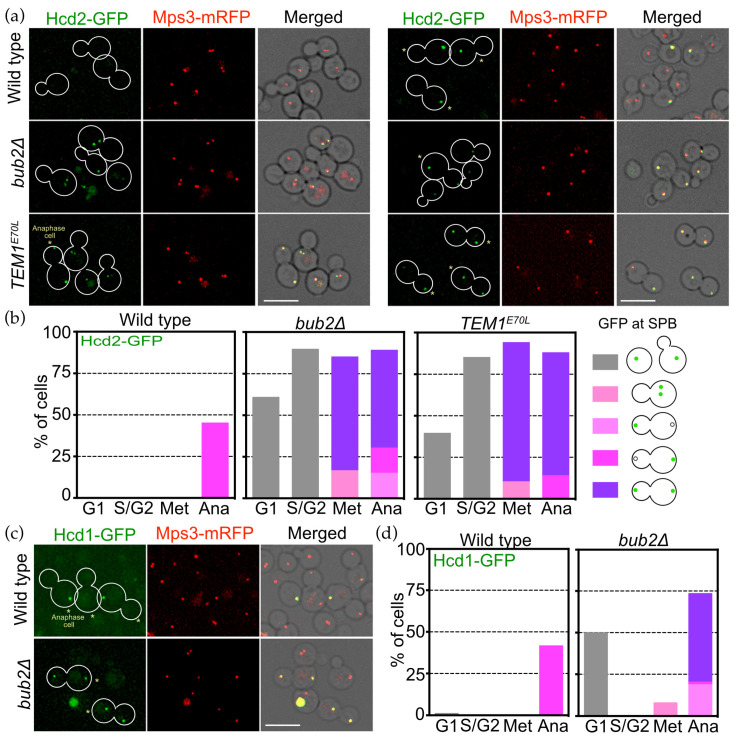
OpBub2 plays an essential role for asymmetric SPB binding of OpHcd2 and OpHcd1. (**a**) SPB localisation of OpHcd2-GFP is regulated by the Tem1 status. Wild type (HPH1605), *bub2*Δ cells (HPH1714), and cells expressing *TEM1^E70L^* (HPH1956)*,* all of which carry *OpHCD2-GFP MPS3-mRFP*, were grown in YPDS medium at 30 °C. In *bub2*Δ and *TEM1^E70L^* cells, Hcd2-GFP appeared at SPBs in metaphase. During anaphase, both the mSPB and dSPB had GFP signal in either *bub2*Δ or *TEM1^E70L^* cells, while only the mSPB had GFP signal in wild type cells. Mps3-mRFP is a marker for the SPB. Shown are deconvolved and projected GFP and RFP images, and merged brightfield, GFP, and RFP images. Scale bar, 5 µm. Asterisks mark late anaphase cells. (**b**) Quantification of the cells in a. (**c**) Asymmetry of Hcd1 SPB localisation depends on the Bub2 function. Wild type cells (HPH1605) or *bub2*Δ cells (HPH1714) that both carry *OpHCD2-GFP MPS3-mRFP*, were grown in YPDS medium at 30 °C. Mps3-mRFP is a marker for the SPB. Shown are deconvolved and projected GFP and RFP images, and merged brightfield, GFP, and RFP images. Scale bar, 5 µm. Asterisks mark late anaphase cells. (**d**) Quantification of the cells in (**c**).

**Figure 8 cells-11-01519-f008:**
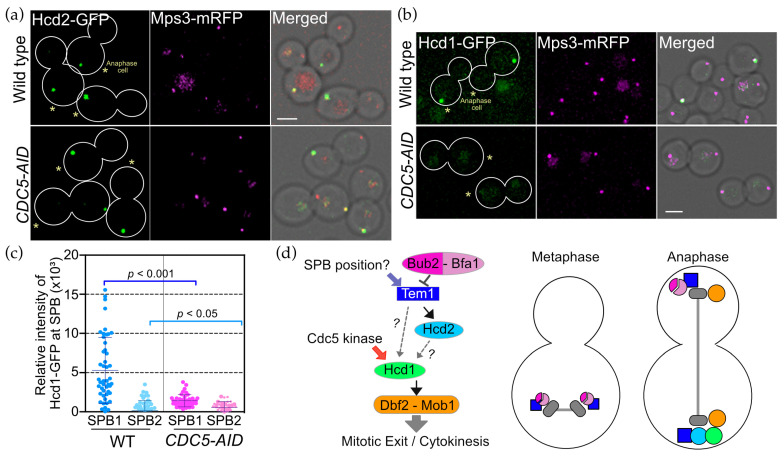
The polo-like kinase is required for SPB binding of OpHcd1. (**a**,**b**) OpHcd1, but not OpHcd2, required OpCdc5 for SPB localisation in anaphase. Wild type and *OpCDC5-AID* cells carrying *OpHCD2-GFP MPS3-mRFP* (HPH1608 and HPH1769), or *OpHCD1-GFP MPS3-mRFP* (HPH1605 and HPH1768) were grown in YPDS medium at 30 °C. IAA was added to a final concentration of 500 µM and incubated for 2 h before capturing images by microscopy. Mps3-mRFP is a marker for the SPB. Shown are deconvolved and projected GFP and RFP images, and merged brightfield, GFP, and RFP images. Scale bar, 2 µm. (**c**) Intensity measurement of GFP at the SPB of anaphase cells in b. The SPB carrying the stronger GFP signal was designated as SPB1 and the other one as SPB2. In total, 48 wild type cells and 41 *OpCDC5-AID* cells were analysed. Statistical significance was determined by the *t*-test. (**d**) Model of the ME-signalling pathway in *O. polymorpha*. See the Discussion for details.

## Data Availability

Data are contained within the article or Appendix A.

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
