# Peer review of "SIN-Like Pathway Kinases Regulate the End of Mitosis in the Methylotrophic Yeast Ogataea polymorpha"

_cells, 2022, doi:10.3390/cells11091519_

Round 1

Reviewer 1 Report

see attached document

Reviewer 2 Report

The MEN pathway is essential for mitosis in budding yeast Saccharomyces cerevisiae. All MEN components are highly conserved in the methylotrophic budding yeast Ogataea polymorpha, except for Cdc15 kinase. Here, this work identified two essential kinases OpHCD1 and OpHCD2, and explored their roles in mitotic exit and cytokinesis. There are several interesting findings in the manuscript. However, the data in this paper is not clearly displayed. The quality of the representative images in the paper needs to be improved.  The statistical comparison needs to be properly displayed. The description of the paper can be more concise and precise.  Please see detailed comments.

1 In figure2A, the authors just showed the representative image of WT type (HPH1047) at 1h. Since the images for hcd1-as (HPH1894) cells at 2h were presented in the same figure. It is necessary to show a representative image of WT at 2h as the proper control. Also, although the size of cells in the three groups looks similar, why the scale bar in these images is not equal?

The statistical comparison in figure2B is confusing. To show the change in cell population in WT and hcd1-as, It is very important to direct compare wt and hcd1-as groups. Is there a significantly different in different cell populations between WT and hcd1-as group?   Figure2b is not really showing anything.

Similar problem in figure3C E, please direct compare wt and the mutant group. The current version is not supporting the conclusion.

Please show a statical comparison for Figure2C. It is hard to get any conclusion without quantification.

The resolution of the images in Figure2D is not good enough to support the authors’ conclusion. Please provide a higher magnification image to prove the point. What is the percentage of cells showing phenotypes like the cells in figure2D?

2 All bar graphs in the paper should be presented with individual data points.

3 The HCD2-as data in Figure3F is the same figure in figure3D. Is there no other examples?

4 In figure4A and figure4B, it is really hard to see any signal of Hcd1-GFP or Hcd2-GFP. Please provide better data. Also, the Mps3-mRFP channel is missing from these two images and the merged data.

5 The figures in the paper are not presented in a consistent format or style. Some figures only show merged channels, some figures show all different channels. Please be consistent. The quality of the images in the paper is not very good and not clear. The author should provide high-quality and convincing images.  

6 From a substantive perspective, the introduction, and results, while providing a very thorough consideration of the present data in light of previous investigations, are very long and should be shortened and the arguments tightened.

Reviewer 3 Report

In this manuscript, Maekawa and colleagues investigated the end of mitosis in methylotrophic yeast Ogataea polymorpha. They identified two essential kinases and described the steps of ME-signaling in this yeast.

The manuscript is straightforward, easy to read, and the experimental strategy logically designed. I recommend this manuscript for publication and have only a few comments to further improve the manuscript:

Comments:

  • In the Material and Method section for some chemicals or Kits the manufacturer and order number are missing.
  • Please proof the figures and improve the quality. In some figures, the lettering appears blurry, making it difficult to read. Other figures have the lettering partially cut off.
  • Fig. 4c: The individual images of the time-lapse are very small. The authors should make it larger so that the statement can be better understood.

Author Response

[Responses to Reviewer#3]

In this manuscript, Maekawa and colleagues investigated the end of mitosis in methylotrophic yeast Ogataea polymorpha. They identified two essential kinases and described the steps of ME-signaling in this yeast.

The manuscript is straightforward, easy to read, and the experimental strategy logically designed. I recommend this manuscript for publication and have only a few

comments to further improve the manuscript:

Thank you for the very supportive comments. We have addressed the points raised by the reviewer. The responses to each comment are as follows:

Comments:

  • In the Material and Method section for some chemicals or Kits the manufacturer and order number are missing.

We have added the manufacturer and order number for all chemicals and Kits in the Material and Method section.

  • Please proof the figures and improve the quality. In some figures, the lettering appears blurry, making it difficult to read. Other figures have the lettering partially cut off.

Thank you for pointing it out. We have proofed all figures and replaced them.

  • 4c: The individual images of the time-lapse are very small. The authors should make it larger so that the statement can be better understood.

We have enlarged the microscopic images in Figure 2-8 to improve the visibility.

Round 2

Reviewer 2 Report

All previous questions have been well addressed. No other questions.